# Mechanisms of Ataxia Telangiectasia Mutated (ATM) Control in the DNA Damage Response to Oxidative Stress, Epigenetic Regulation, and Persistent Innate Immune Suppression Following Sepsis

**DOI:** 10.3390/antiox10071146

**Published:** 2021-07-20

**Authors:** Laura A. Huff, Shan Yan, Mark G. Clemens

**Affiliations:** Department of Biological Sciences, University of North Carolina at Charlotte, Charlotte, NC 28223, USA; Laura_Huff@med.unc.edu (L.A.H.); shan.yan@uncc.edu (S.Y.)

**Keywords:** ataxia telangiectasia mutated (ATM), DNA damage response, innate immune response, oxidative stress, epigenetic regulation, p53, sepsis

## Abstract

Cells have evolved extensive signaling mechanisms to maintain redox homeostasis. While basal levels of oxidants are critical for normal signaling, a tipping point is reached when the level of oxidant species exceed cellular antioxidant capabilities. Myriad pathological conditions are characterized by elevated oxidative stress, which can cause alterations in cellular operations and damage to cellular components including nucleic acids. Maintenance of nuclear chromatin are critically important for host survival and eukaryotic organisms possess an elaborately orchestrated response to initiate repair of such DNA damage. Recent evidence indicates links between the cellular antioxidant response, the DNA damage response (DDR), and the epigenetic status of the cell under conditions of elevated oxidative stress. In this emerging model, the cellular response to excessive oxidants may include redox sensors that regulate both the DDR and an orchestrated change to the epigenome in a tightly controlled program that both protects and regulates the nuclear genome. Herein we use sepsis as a model of an inflammatory pathophysiological condition that results in elevated oxidative stress, upregulation of the DDR, and epigenetic reprogramming of hematopoietic stem cells (HSCs) to discuss new evidence for interplay between the antioxidant response, the DNA damage response, and epigenetic status.

## 1. Introduction

It is well known that low basal levels of reactive oxygen and nitrogen species (ROS and RNS, respectively, or RONS) play crucial signaling roles within cells under normal physiological conditions. The major ROS involved in signaling include the superoxide radical (O2^−•^), hydrogen peroxide (H_2_O_2_), formed by reduction of superoxide via superoxide dismutase (SOD) and spontaneous dismutation of superoxide, and the hydroxyl radical (HO^.^) formed by H_2_O_2_ in the presence of free transition metals. RNS include the nitric oxide (NO) radical and peroxynitrite (ONOO^−^), the product of a chemical reaction between O2^−•^ and NO [1]. Controlled subcellular localization and accumulation of oxidants allows for use of these reactive species as signaling molecules [2].

While low physiological levels of RONS are crucial for cell signaling and growth, levels that exceed a critical threshold of antioxidant capacity within the cell can lead to damage of cellular biomolecules including nucleic acids, proteins, and lipids. Recent studies have revealed a concentration-dependent response within the cell to different levels of RONS. Three levels of cellular response to oxidants are identified as (i) eustress at normal physiological conditions (i.e., H_2_O_2_ concentrations of 1–10 nM), (ii) adaptive stress via redox sensors and effectors such as the Keap1-Nrf2 and JNK-NF-κB transcription factor activation pathways (i.e., H_2_O_2_ concentrations of 10–100 nM), and (iii) excessive oxidative stress leading to cellular distress and damage to cellular components (i.e., H_2_O_2_ concentrations > 100 nM) [3]. Thus, under elevations in RONS slightly above physiological levels, adaptive cellular antioxidant signaling pathways are capable of responding to oxidative stress up to a threshold level. However, excessive production and accumulation of RONS occurs and is implicated in a broad host of pathological states including neurodegenerative diseases such as Alzheimer’s [4], Parkinson’s, and Huntington’s diseases [5], atherosclerosis [6], rheumatoid arthritis [7], ischemia reperfusion injury [8], aging and age-related conditions such as cardiovascular diseases, chronic kidney disease [9], cancer [10], and sepsis [11].

Control of excessive oxidative stress is vital within cells to maintain cellular and genome integrity. Damage to the genome is particularly detrimental to host organisms and cells have evolved complex pathways to detect and coordinate response to and repair of DNA damage. Studies indicate that many pathophysiological states that are characterized by elevated oxidative stress are also associated with upregulation of the DNA damage response (DDR). For example, markers of DDR upregulation are found in patients of inflammatory diseases such as cancer [12,13] and chronic systemic autoimmune diseases [14]. Acute parasitic infection with *Trypanosoma cruzi* was found to induce phosphorylation of histone H2AX (i.e., γH2AX) [15], migraine headaches are associated with increased marker of oxidative DNA damage 8-hydroxy-2′-deoxyguanosine (8-OHdG) [16], and patients with sepsis were found to have increased markers of oxidative DNA damage (8-OHdG) levels compared to healthy controls [17]. Certain pathophysiological conditions associated with elevated RONS are also correlated with epigenetic changes to the nuclear chromatin of affected (i.e., oxidative stress) cells. One striking example are the epigenetic changes in macrophages and dendritic cells during acute inflammatory conditions such as sepsis that are responsible for suppression of these cells and hypo-responsiveness upon subsequent infection in sepsis survivors. Lasting changes, as in the latter example, can only occur when epigenetic changes are made to stem cells that replenish the supply of successor cells throughout the lifetime of an individual.

Sepsis is a prime representative inflammatory disease characterized by high oxidative stress through which epigenetic reprogramming of hematopoietic stem cells (HSCs) by pathways putatively upregulated by RONS may be explored. Recent evidence suggests that DDR factors such as ATM, p53, and p21 may be involved in initiating these epigenetic changes. Here we discuss the newly emerging evidence for interplay between the antioxidant response, the DNA damage response, and the epigenetic regulation, in particular of hematopoietic stem cells. Implications from this connectiveness may open new avenues of mechanistic research for a host of different pathologies associated with elevated oxidative stress, including cancer, autoimmune disease, and aging.

## 2. Physiological Regulation and Response to RONS

### 2.1. Antioxidant Response to Elevated RONS

Basal levels of RONS have been demonstrated to exhibit crucial signaling roles within the cell. For example, ROS such as H_2_O_2_ can directly oxidize protein cysteine residues leading to signaling cascades important in processes such as differentiation and proliferation [18,19,20,21,22], and gradients of H_2_O_2_ are used as signals to coordinate leukocyte migration to sites of inflammation for wound repair [23]. The superoxide radical (O_2_^−^) can activate the Ras/Raf/MAPK pathway and change gene expression patterns involved in growth and proliferation [1,22] and in response to angiotensin II in the central nervous system, leads to increased vasopressin secretion and sympathetic outflow [1]. Additionally, mitochondria can function as danger sensors and upregulate production of ROS as a signaling mediator [24]. The RNS NO is important in neuronal, endothelial, and immune cell signaling, where it is involved in a diverse array of signaling pathways including neurotransmission, inflammatory immune responses, modulation of ion channels, vascular homeostasis, penile erection, bladder control, lung vasodilation, and peristalsis, dependent upon cell and tissue type, concentration, and localization [25].

Under basal and minimally elevated RONS conditions, the cellular antioxidant response is capable of controlling the redox level within the cell and maintaining redox homeostasis. One of the primary defense mechanisms to oxidants is through transcriptional upregulation of antioxidant response elements (ARE), cis-acting enhancer elements located in the promoters of detoxification enzyme genes such as GSTA2 (glutathione S-transferase A2) and NQO1 (NADPH: quinone oxidoreductase 1) in response to a variety of stress signals. One of the major regulators of ARE activated gene upregulation is the Kelch-like ECH-associated protein 1 (Keap1)-Nrf2 pathway that upregulates of antioxidant and detoxification gene [26,27]. Endogenous antioxidants include enzymes such as superoxide dismutase (SOD), catalase (CAT), glutathione peroxidase (GPx), DT-diaphorase, and non-enzymatic compounds including albumin and bilirubin. Under minimal elevation of oxidative species, the cellular antioxidant response is able to maintain redox homeostasis; however, when RONS elevation exceeds the antioxidant capacity of the cell, oxidative stress accumulates and the RONS can cause extensive damage to biomolecules, leading to either cell death or upregulation of cellular maintenance and repair pathways [28].

### 2.2. Sources of ROS and RNS in Pathophysiological States

In pathological states associated with chronically elevated oxidative stress, RONS are produced by a combination of sources including mitochondria, NADPH oxidases (NOX), nitric oxide synthases (NOS), peroxisomes, and additional specific enzymes such as xanthine oxidase, cyclooxygenases, cytochrome p450 enzymes, and lipoxygenases. Evidence suggests that ROS such as superoxide and H_2_O_2_ are produced by mitochondria primarily at complex I or complex III of the electron transport chain. NADPH oxidases were once considered exclusive to phagocytic cells but are now known to comprise a family of seven NOX NADPH members, Nox1–5 and Duox1–2, that are expressed in the plasma membrane of a broad range of cell types and widely distributed in various tissues [29]. Rapid production of superoxide by NADPH oxidase upon activation by proinflammatory signals comprises the oxidative burst observed in phagocytic cells for the purpose of pathogen killing. Superoxide generated extracellularly may constitute part of the oxidative environment within phagolysosomes or may also undergo spontaneous dismutation outside of the cell to H_2_O_2_ and diffusion back across the plasma membrane directly or through aquaporin channels [30].

NO is produced from L-arginine by a family of NOS enzymes (iNOS, eNOS, and nNOS) that are expressed in specific tissues and by different regulatory mechanisms. While low levels of NO produced by constitutively expressed eNOS and nNOS are es-essential for blood flow regulation and neurotransmission, NO is produced in much larger quantities by inducible NOS (iNOS) in innate immune cells such as neutrophils, macrophages, and dendritic cells in a transcriptionally regulated and calcium-insensitive manner. Upregulation of iNOS and subsequent production of NO is induced by pro-inflammatory cytokine signaling (i.e., TNFα, IL-1β, IFNγ) and activation of pattern recognition receptors (PRR) such as the toll-like receptors (TLR) [25]. NADPH oxidase is the primary source of superoxide radical species and H_2_O_2_ while iNOS is the major source of NO under proinflammatory conditions. These ROS and RNS species, superoxide and NO, can undergo spontaneous chemical reaction to yield peroxynitrite (ONOO^−^).

### 2.3. Activation of the DNA Damage Response (DDR) by RONS

Maintenance of the nuclear genome is vital for survival and eukaryotic organisms have evolved an elaborate array of pathways to detect and coordinate repair of various forms of DNA damage [31]. DNA lesions can generally occur through a wide range of sources, including UV irradiation and genotoxic agents, but are primarily the result of endogenously generated RONS under conditions of proinflammatory responses. Different types of DNA damage result in varied repair pathways and include alterations to bases, formation of bulky adducts, mismatched base pairs, crosslinking, loss of a base (also known as AP site), single-strand DNA breaks (SSBs), and double-strand DNA breaks (DSBs).

Repair pathways can be broadly divided into SSB and DSB repair. DSBs are considerably more deleterious to the host organism due to the possibility of genetic loss and genome rearrangements. Cells that are in the S or G2 phase of the cell cycle can undergo the more robust form of DSB repair, homologous recombination (HR), using the sister chromatid as a template for repair. Cells in all other phases (i.e., quiescent or in the G1 phase) utilize the HR pathway with the homologous chromosome or the less robust non-homologous end-joining (NHEJ) repair pathway, which directly ligates the two broken ends together and can result in short nucleotide deletions [32]. Sensors of DNA damage interact with chromatin to detect the damage and recruit additional factors that initiate a signaling cascade to downstream transducer and effector proteins. Briefly, SSBs result in recruitment and activation of the phosphatidylinositol 3-kinase (PI3K)-related kinase (PIKK) ataxia telangiectasia and Rad3 related (ATR), and subsequent activation of kinase Chk1. DSBs result in recruitment and activation (i.e., phosphorylation) of another PIKK ataxia telangiectasia mutated (ATM) and subsequent activation of kinase Chk2 [33].

The canonical function of ATM as a master regulator of the DSB DDR pathway is well-established. In this role, ATM mediates S-phase checkpoint activation in conjunction with initiation of DDR signaling [34]. In most cell types ATM is predominantly localized to the nucleus as an inactive and noncovalent homodimer [35]. Upon detection of DSBs, human ATM is acetylated at lysine K3016 (in the FATC domain), autophosphorylated at serine S1981, and subsequently monomerized and may also be recruited to sites of DNA damage by the Mre11, Rad50, Nbs1 (MRN) protein complex at the DSB site [36]. Autophosphorylation may not be a requirement for activation in response to DNA damage in non-human species such as mice or *Xenopus* [37,38,39]. Thus, in response to DSBs, ATM is converted into an active monomer form that phosphorylates an estimated hundreds of substrates that are involved in cell cycle checkpoints, DNA repair, and additional cell responses [34,35,40]. Following response to DNA damage, ATM phosphorylates and activates p53 and Chk2, initiating DNA repair, cell cycle arrest, and other processes. Under such conditions, the phosphorylated ATM forms discrete foci in the nucleus. ATM and Chk2 can both lead to γH2AX, which is a commonly used marker of DSBs. H2AX is phosphorylated at serine S139 by ATM. DSB foci uniquely contain γH2AX, which spreads throughout neighboring chromatin domains over more than 1 Mb on each side, nucleating additional response sites to amplify the repair signal [41,42].

Phosphorylation of p53 at serine Ser20, a downstream effector transcription factor, leads to its activation, translocation to the nucleus, and transcriptional upregulation of genes that regulate the cell cycle, DNA repair, apoptosis, senescence, and angiogenesis, and the ubiquitin ligase MDM2, leading to accumulation of p53 [43]. One of the many downstream targets of activated p53 transcriptional upregulation is p21^Cip1/Waf1^ (p21), a cyclin-dependent kinase inhibitor that primarily inhibits cyclin/cyclin-dependent kinase (CDK) complexes and regulates the G1/S phase cell cycle progression. Upregulation of p21 is one of the key mechanisms through which p53 mediates cell cycle arrest in response to DNA damage detection [44].

### 2.4. Involvement of the DDR in Epigenetic Modifications

Epigenetic modification of chromatin, including modifications to chromatin chemical markers and structure that result in changes of gene expression without modification to the DNA sequence, can facilitate DNA repair processes [45]. Modifications including methylation of DNA, acetylation of histones, chromatin remodeling, and nucleosome removal and replacement are imperative for repair processes. Epigenetic modification is currently most evident during and after DSB repair [46]. In particular, lasting changes in the methylation profile of DNA have been shown to occur at the site of DSBs repaired by the HR pathway [32]. DNA damage response effector proteins are known to interact with epigenetic regulators such as the SWI-SNF (SWItch/Sucrose Non-Fermentable) nucleosome remodeling complex, DNA methyltransferases (DNMT), ten-eleven translocation dioxygenases (TETs), histone deacetylases (HDACs), histone acetylase enzymes (HATs), etc. These multiple pathways link the DDR to epigenetic regulation.

### 2.5. RONS can Directly Participate in Chemical Modification of Chromatin

Evidence suggests that RONS can participate in epigenetic modifications to DNA and histones via direct and indirect mechanisms—this was reviewed extensively by T. Kietzmann et al. recently (2017), and will be briefly discussed here [47]. Firstly, ROS can directly modify nucleotides. For example, hydroxyl radicals can abstract hydrogen atoms from the methyl group of 5mC leading to formation of 5hmC, or ROS can directly oxidize guanosine to 8-oxo-2′-deoxyguanosine (8-oxodG), which can both indirectly lead to demethylation or hypomethylation, respectively. Additionally, superoxide has been suggested to directly participate in DNA methylation of cytosine through deprotonation of cytosine or through neutralization of the positive charge on the sulfur atom of S-adenosyl-L-methionine (SAM), an endogenous methyl donor, followed by transfer of a methyl group from S-adenosyl-L-methionine (SAM) [48,49]. Peroxynitrite can induce nitration and oxidation of histones H1, H2B, and H3, leading to chromatin structural rearrangements [50]. Histones can also undergo oxidation by RONS. ROS can directly oxidize arginine and lysine residues in histone H3, which may again lead to chromatin structural rearrangement and differential regulation. The Cys110 residue of histone H3 can additionally be S-glutathionylated, which leads to opening of the chromatin structure [51]. Additionally, lipid peroxidation products that can form in the presence of oxidative species (e.g., 4-oxo-2-nonenal) can produce lysine adducts on histones H2, H3, and H4 [52].

Interestingly, evidence suggests that ROS does not necessarily mediate a global change in DNA methylation patterns. Rather, differential expression patterns are observed under conditions of increased ROS that may indicate localized trends. For example, elevated levels of ROS have been shown to globally increase levels of DNA methylation in conditions associated with hypoxia/reoxygenation [53,54,55]. However, in separate studies, elevated ROS (superoxide) has been shown to cause DNA demethylation [56,57]. In general, it is evident that although the precise mechanisms are yet to be elucidated, changes in RONS levels affect chromatin structure and the epiregulome.

RONS can also indirectly affect DNA and histone modification. For example, RONS is known to affect the activity of DNMT enzymes; however, its role in modulating DNMT activity appears contradictory and may depend on local subcellular conditions. For example, ROS can either increase or decrease the activity of DNMTs, through the upregulation of DNMT expression by activation of HIF1α or reducing the availability of cofactor SAM, respectively. ROS can also increase the activity of TET proteins, leading to increased occurrence of demethylation due to increased 5hmC abundance [56,58,59]. Chronic elevation of H_2_O_2_ has been shown to cause recruitment of DNMT1 to damaged DNA sites, forming an epigenetic silencing complex that leads to hypermethylation and silencing of the complexed gene [60]. ROS can modulate histone methylation, including activating marks (i.e., H3K4me2/3) and repressive marks (i.e., H3K9me2/3 and H3K27me3) [61,62]. RONS can modulate histone methylation by altering the activity and expression of histone methyltransferases (HMTs) and histone demethylases (HDMs). RONS have also been reported to modify histone acetylation through increasing the activity of HATs, and through posttranslational modifications (i.e., S-glutathionylation, S-nitrosylation, acetylation, and phosphorylation) to HDAC proteins, resulting in reduction of HDAC function and increased histone acetylation [47].

Overall, the evidence indicates that RONS play crucial roles in regulation of epigenetic modifications to nuclear DNA through redox mediators, direct utilization as cofactors in chemical modifications on DNA, and through regulation of epigenetically modifying enzyme expression.

## 3. Pathological States Associated with High Oxidants Result in Epigenetic Changes to Stem Cells

### 3.1. Sepsis as a Model for Elevated Oxidative Stress and Epigenetic Modifications

Sepsis is a prime example of a proinflammatory pathological condition characterized by excessive oxidative stress with resultant epigenetic modifications to innate immune cells and hematopoietic stem cells (HSCs). Sepsis is defined as a state of life-threatening organ dysfunction that occurs due to dysregulated and excessive host response to an infection that presents with a wide array of symptoms such as hypoxia, hypotension, hypercoagulation, circulatory failure, tachycardia, high blood lactate levels, metabolic disorders, and multiple organ failure (MOF) [63,64]. Thanks to advances in emergency medical care the number of patients who survive sepsis has risen steeply in recent years. However, a large portion of sepsis survivors face long-term complications including physical, psychological, and cognitive disorders, as well as persistent immunological impairment and immunosuppression that lead to a lifelong increased risk of mortality. The reasons for this increased rate of mortality among sepsis survivors is thought to be due to changes in their innate immune system function, resulting in immune dysregulation and inability to respond adequately to subsequent infection. Dysregulation of the innate immune function in sepsis survivors is termed the persistent inflammation, immunosuppression, and catabolism syndrome (PICS). The mechanisms underlying PICS remain unclear and are the subject of intensive research [65].

The emerging picture of the pathophysiology of sepsis is complex and differs between individuals. Generally, initial responses of the innate immune system to pathogen-associated molecular patterns (PAMPs) and danger-associated molecular patterns (DAMPs) which signal the presence of infectious agents, trigger production of cytokines such as type-I interferons (IFNα/β), IL-1β, and IL-6. These cytokines initiate a systemic signaling cascade that result in further activation of immune cells and circulation of additional cytokines, such as type-III IFN (IFNγ), often referred to as a “cytokine storm”, as depicted in Figure 1. Through activation of NFkB and JAK/STAT1 signaling pathways, proinflammatory mediators are upregulated, leading to production of RONS, oxidative stress from infiltrating inflammatory cells. Within hours a compensatory hypo-inflammatory response is mounted to initiate tissue repair [66]. The hyper-inflammatory phase is characterized by increased oxygen consumption, elevated ATP production, a metabolic switch to aerobic glycolysis (the Warburg effect), increased catabolism, and up-regulation of NOX, iNOS, and antimicrobial RONS. These responses occur in both inflammatory cells, especially neutrophils and macrophages, as well as in vascular and somatic parenchymal cells. As a result, oxidative and nitrosative stress are important components of cell injury in sepsis. In contrast, the hypo-inflammatory response consists of decreased oxygen consumption, mitochondrial respiration, fatty acid oxidation (FAO), and tolerance of immune cells to pathogenic threats [67,68]. Recent research indicates that these two processes (hyper and hypo-inflammation) may occur simultaneously [69]. After recovery, however, the physiological symptoms of sepsis survivors are typically that of immuno-impairment [70].

Recent findings suggest that the immunosuppression of innate immune cells in sepsis survivors is a form of innate immune memory, which is defined as a functional reprogramming of innate immune cells after a pathogenic encounter leading to either an enhanced (trained) or reduced (tolerant) response to subsequent encounters [72]. Macrophages, along with neutrophils, dendritic cells, and T-helper (TH) cells, are a major component of the immune response to sepsis and are one of the cell types most impacted by immunosuppression due to their critical role in the immune response upon subsequent infection. Phenotypic differences between macrophages from healthy individuals and those from sepsis survivors with impaired immune function (PICS) have been studied extensively. Genes with inducible expression in macrophages under normal physiological conditions that are not inducible in ‘tolerant’ (Class T) macrophages include genes such as iNOS (NOS2), CD40, IL-6, IL-1β, caspase 12, etc. [73].

Macrophages can assume an array of phenotypes depending upon the stimulus they are exposed to, similarly to the polarizability of TH cells into Th1 or Th2 phenotypes. Macrophages can be generally polarized into one of two extremes along a phenotypic gradient: either a pro-inflammatory M1 (classically activated) or anti-inflammatory M2 (alternatively activated) phenotype [74]. Intriguingly, the genes that are not inducible in tolerant (T) macrophages are those associated with the M1 phenotype. Thus, numerous studies indicate that macrophages of sepsis survivors cannot assume the M1 phenotype upon stimulation with novel pathogen exposure or with a typical pattern recognition receptor (PRR) stimulant such as lipopolysaccharide (LPS).

The mechanisms of macrophage tolerance and the associated epigenetic changes that occur are the subject of current research and remain to be fully elucidated. Studies from various groups isolating and investigating downstream signaling from different PRRs indicate that multiple cytokine signaling pathways can activate the myelosuppressive state. One commonly used stimulant is LPS, a major activator of toll-like receptor 4 (TLR4) on macrophages. Activation of TLR4 leads to the same phenotypic changes in macrophages in vitro as detected in septic survivors and TLR4 activation in macrophages is also known to increase RONS production [73,75]. However, the distinct phenotypic changes indicative of septic myelosuppression can also be initiated through other pathogen-associated molecular pattern (PAMP) signaling pathways as well, and are therefore not exclusive to chronic TLR activation [65] The pathogenesis of sepsis is highly complex and various signaling pathways have been implicated in the observed myeloid cell reprogramming.

### 3.2. Evidence for Epigenetic Changes to Innate Immune Cells

Evidence is rapidly mounting that epigenetic modifications such as chromatin remodeling, DNA methylation, noncoding RNAs, and histone methylation and acetylation play a major role in the induction of the myelosuppressive state in sepsis survivors [65,73,76,77,78,79,80,81]. For example, one study has shown that LPS stimulation can result in transcriptional silencing of several pro-inflammatory genes through the action of miRNAs [78]. Another showed that genes induced upon secondary stimulation in tolerant murine macrophages fall into two distinct categories: pro-inflammatory and anti-microbial/metabolic. Epigenetic acetylation and methylation of histone 3 was shown to differentially regulate these changes [73]. Another key study showed that precise and selective chromatin modification at promotor regions of inflammatory genes occurs in monocytes of human sepsis donors [82]. Lastly, two key studies demonstrated that hypo-responsiveness of the iNOS gene during myelosuppression is due to hypermethylation of CpG nucleotides and H3K9me methylation [83], and that demethylation of certain NF-*κ*B responsive enhancer elements are associated with transactivation of iNOS [84]. These studies indicate that epigenetic modifications do occur during sepsis and can result in phenotypic changes such as suppression of iNOS activation and cytokines in macrophages, which are hallmark traits of myelosuppressed cells.

### 3.3. Evidence for Epigenetic Changes to HSCs from Sepsis

Innate immune cells present during the acute phase of sepsis undergo chromatin modifications that alter their cellular function; however, the permanent change seen in immune cells of sepsis survivors for years after the initial insult and well beyond the lifespan of any innate immune cells present at the time is due to changes to the hematopoietic stem cell (HSC) pool that give rise to all successive innate immune cells. Numerous studies indicate that induction of emergency hematopoiesis, the rapid activation and differentiation of HSCs within the bone marrow (BM) environment, and even depletion of HSCs occur during sepsis [85,86,87]. In addition, recent studies provide evidence that during the acute phase of sepsis, epigenetic changes are made to HSC nuclear chromatin [88]. This critically important piece of the emerging picture provides a mechanism for how innate immune cells of sepsis survivors remain permanently suppressed. Changes to gene transcriptional programs within mature innate immune cells during sepsis last only as long as the reprogrammed cells. In contrast, changes to the epigenetic status of HSCs can be imparted to all progeny immune cells.

In a murine model of LPS TLR4 activation-induced sepsis, signaling through the downstream adaptors MYD88 and TRIF result in permanent alterations to the HSC transcriptional programs. More specifically, MYD88 activation is a key cause of myelosuppression during sepsis, while TRIF activation has a greater effect on HSCs. Taken together, it was shown that signaling through both MYD88 and TRIF contribute to the permanent alteration of the transcriptional programs of HSCs [89]. In another murine model of sepsis, HSCs from septic mice were significantly impaired in their ability to self-renew and repopulate and also to produce myeloid and granulocyte-monocyte progenitor cells [90]. Interestingly, mature macrophages are also known to interact with the HSC pool and promote HSC expansion and differentiation under inflammatory conditions [91]. This suggests a potential role for activated macrophages in the induction of epigenetic suppression of HSCs during sepsis.

### 3.4. Evidence of DNA Damage from Sepsis

While it is broadly accepted that RONS are capable of inducing DNA damage and triggering upregulation of the DDR [92], evidence has accumulated that RONS cause DNA damage in innate immune cells specifically during sepsis. In a study of human sepsis patients, the level of oxidative DNA damage, measured by 8-OHdG levels, found that DNA damage was increased in septic patients compared to healthy controls [17]. These increases in DNA damage were suggested to be correlated with increased levels of oxidative stress (i.e., RONS).

Additional studies focused on determining the source of RONS that give rise to DNA damage found that cellular responses downstream of iNOS induction may be responsible during inflammatory conditions such as sepsis. One study showed that bone marrow-derived macrophages (BMDMs) treated with an iNOS inhibitor (AGHS) and with and the M1 phenotype-stimulating LPS + IFNγ combination, a common indicator of DNA damage, γH2AX, was prevented [93]. It was also found that type I IFN receptor signaling is necessary for γH2AX phosphorylation in BMDMs infected with LM. The results of this study overall indicate that both iNOS upregulation and type I IFN signaling are important for induction of DNA damage in proinflammatory activation of BMDMs.

## 4. The Upregulation of the DDR and the Epigenetic Modifications to Stem Cells May Be Causally Linked

### 4.1. Dual Roles of ATM as a Master Regulator of the DDR and a Redox Sensor

Loss of ATM function results in the autosomal recessive disorder ataxia-telangiectasia (A-T), which causes ataxia due to cerebellar neuron cell loss and telangiectasia [34]. A-T is characterized by accumulation of RONS [94,95], high oxidative stress, loss of hematopoietic stem cells, premature aging, immunodeficiency, extreme sensitivity to ionizing radiation, chromosomal instability, defective spermatogenesis, and increased risk for certain cancers, especially lymphoid cancers [34,96]. Experimental deletion of ATM renders aspects of the antioxidant response non-functional, which results in accumulation of ROS and damage to various cellular structures, including DNA [34]. Dysregulation of the antioxidant response and immune signaling upon loss of ATM suggest a role for ATM in these critical cellular pathways.

ATM has at least two known and disparate functions: (i) as a master coordinator of the DDR and (ii) as a redox sensor [97], summarized in Figure 2, and the mechanism by which ATM is activated is different for each stimulus. Activation via the DDR requires the inactive ATM non-covalent homodimer located in the nucleus to be recruited to an MRN complex, acetylated at lysine K3016, undergo autophosphorylation at serine S1981 (in human cells), and monomerize. Activation via oxidation by RONS does not require phosphorylation or monomerization. Instead, cysteine residues on ATM are directly oxidized by RONS (such as H_2_O_2_) forming a disulfide bridge between two ATM monomers and resulting in a covalently linked homodimer. One disulfide bridging cysteine residue in particular (the C2991 residue in humans) is required for activation of ATM in this manner. Additionally, the set of substrates targeted by ATM are different for each stimulus, resulting in different downstream signaling pathways [98]. Of the large number of identified ATM substrates, it is currently unknown whether any are specific to the oxidative activation pathway. Indeed, the function of most ATM targets currently remains unknown [34].

Interestingly, the DNA damage and direct oxidation signaling pathways of ATM are known to function independently of one another [99]. Inactivation of covalent disulfide bridging of the monomers does not disturb MRN activation of the enzyme [100,101,102]. In another study, ATM autophosphorylation and activation was found to occur upon treatment with low levels of hydrogen peroxide that did not result in DNA damage (as indicated by γH2AX staining) in primary and transformed human cells [103]. Additionally, covalently linked ATM homodimers are also able to bind the MRN complex and upregulate the DSB DDR. This makes sense evolutionarily since stressful conditions will likely result in elevated ROS as well as DNA damage occurring simultaneously [34].

The mechanism through which ATM senses oxidative stress, through oxidation of cysteine residues and formation of a disulfide bridge between monomers bears remarkable similarity to mechanisms through which other redox sensors detect RONS, such as glutathione; however, these signaling pathways through which ATM functions as a redox sensor remain yet to be fully elucidated. ATM has been shown to perform diverse functions as a redox sensor as well as a regulator of the DDR, placing it in a role that could directly connect the redox regulatory pathways and antioxidant response of the cell to the DDR.

### 4.2. ATM as a Regulator of Epigenetic Change

ATM has been demonstrated to modulate the activity of several transcription factors (TFs) with known epigenetic regulatory functions such as HIF1α and NFκB, providing a link between ATM and epigenetic regulation. Interestingly, ATM activated by oxidative stress may activate HIF1α through two separate routes: (i) through activation of AKT and mTORC1 and (ii) through direct activation of HIF1α. Interaction of ATM with downstream mTORC1 and HIF1α present myriad routes through which ATM could regulate epigenetic control.

When activated via oxidation and covalent homodimerization, ATM activates AKT, a known key activator of mammalian target of rapamycin (mTOR) complex 1 (mTORC1). The mTORC1 complex is a major regulator of metabolism, proliferation, growth, autophagy, and anabolic processes within the cell. mTOR is a PIKK and a master regulator of cellular homeostasis [34,104]. One downstream target of mTORC1 is the transcription factor HIF1α. Oxidized ATM activates AKT, thereby increasing translocation of GLUT4 receptors to the plasma membrane and increasing uptake of glucose while upregulating glucose-6-phosphate dehydrogenase (G6PD) and also resulting in activation of HIF1α [99,105]. An alternative and more direct activation of HIF1α by ATM has been described, where ATM that is activated by hypoxic stress directly phosphorylates HIF1α on serine S696 [106]. HIF1α is known to modulate various forms of epigenetic regulation including miRNA expression, histone modification, and chromatin structure [107]. Additionally, HIF1α modulates gene activity of epigenetic regulators histone lysine demethylases (including K-specific demethylases or Jumonji C lysine demethylases) [107]. Activation of this TF, that is tightly associated with epigenetic change, could be one mechanistic pathway through which ATM modulates epigenetic reprograming of innate immune cells and HSCs during sepsis.

Activation of mTORC1 is also known to activate effectors that interact with epigenetic regulators required for modifying chromatin structure and function to control gene expression [108]. For example, mTORC1 signals to downstream epigenetic effects such as regulators of ribosomal gene transcription. mTORC1 can additionally modulate binding of HATs to specific gene promotors, thereby altering gene expression profiles. For example, in yeast mTORC1 promotes binding of the Esa1 HAT to RP gene promotors. Inhibition of mTORC1, such as by rapamycin, also decreases histone H4 acetylation at these promotors by initiating release of Esa1, reducing transcription of the RP gene [109]. mTORC1 inhibition may also increase the activity of specific HDACs, such as Rpd3 in yeast to transcriptionally repress ribosome biogenesis [110].

The transcription factor NFκB is required for activation of the iNOS and proinflammatory cytokine TNFα genes. Interestingly, ATM has been shown to be required for NFκB activation through certain PRR-mediated pathways and more broadly in response to genotoxic and oxidative stress via post-translational modifications [111]. One study demonstrated that ATM is a key mediator of TLR3 and NOD2-induced activation of NF-κB [112]. In a genome-wide RNAi screen, ATM was found to undergo phosphorylation and interact with IκB kinase (IKK) complex proteins (TAK1, NEMO, IKKα, and IKKβ), resulting in activation of NFκB and positive modulation of TLR3 signaling. The authors note that it is currently unknown whether the same subunits are required for NFκB gene transcription upregulated by other PRRs, such as TLR4. TAK1 (TGFβ-activated kinase 1) is an activator of IKK. In this study, ATM was shown to be essential for phosphorylation of Ikβ and translocation of NFκB to the nucleus, resulting in binding of NFκB to response elements and modulation of gene transcription [112]. ATM is therefore known to regulate the innate immune response of TLR3 -mediated activation of NFκB with the potential for its involvement in NFκB activation via other PRRs remaining an open question.

Studies indicate that ATM is additionally involved in the NF-κB-mediated cellular senescence, stem cell dysfunction, and aging, as demonstrated in senescent murine Ercc1-/- MEFs [111]. Key markers of the senescence-associated secretory phenotype (SASP) include IL-6, IL-1α, IL-1β, and TNFα, which can disrupt stem cell quiescence and induce secondary senescence. Cellular senescence is known to be mediated by p53/p21 and p16^INK4a^/retinoblastoma (Rb) tumor suppressor pathways in response to stressors including DNA damage and oxidative as well as inflammatory stress [111].

### 4.3. NFκB as an Epigenetic Modifier

The results of several studies indicate that NFκB can induce epigenetic modifications. For example, NFκB has been shown to suppress levels of specific histone methylation marks. In one study NFκB repressed expression of histone H3K36 trimethylases NSD1 and SETD2 and concurrently lowered levels of H3K36me3 markers. Conversely, genomic and pharmacological inhibition of NFκB (using MDA-MB-231 breast cancer cells overexpressing IκBαSR and P65^−/−^ MEFs and DMAPT) results in elevated H3K36me3 histone methylation [113]. NFκB has also been shown to induce gene expression of histone demethylases KDM6B (demethylates H3K27), PHF2 (demethylates H4K20), and KDM2B (demethylates H3K36me1, H3K36me2, and H3K4me3) [114,115,116]. Additionally, the NFκB subunit p65 associates with the transcription start site (TSS) of the up- and downregulated miRNA genes and is associated with changes in histone methylation (H3K27me3 and H3K4me3) upon infection of B lymphocytes with Epstein-Barr virus (EBV) [117]. In HCT116 cells lacking DNA methyltransferases (DNMT), TNFα-induced activation of NFκB was blocked and expression of IκBa was higher than in wild type cells. Higher IκBa expression was correlated with significant decreases in the epigenetic methylation markers in CpG dinucleotides at IκBa promoters. Additionally, increased levels of modified histones were found to be associated with the IκBa promotor in DNMT knock-out cells, including H3K14ac, H3K4me2, and H3K9me3, suggesting modified chromatin structure at the IκBa locus [118]. Altogether, these data indicate that one common function of NFκB activation is histone demethylation, although its function has been shown to be differentially regulated in a cell type or other conditionally dependent manner.

### 4.4. Epigenetic Regulation of iNOS Gene

One downstream target of proinflammatory NFκB activation is the iNOS gene, NOS2. NFκB upregulates expression of iNOS in innate immune cells in response to cellular stress signals, such as proinflammatory cytokines and PAMPs that act on PRRs during sepsis. Epigenetic control of the NOS2 gene in myelosuppressed cells via hypermethylation of the promotor and histone H3K9me methylation has been demonstrated experimentally [82,83] and CpG methylation of the NOS2A promotor has been shown to control inducibility of this gene. Specifically, increased methylation is correlated with decreased ability of NOS2 activators to upregulate its expression [83]. Increases in methylation of the NOS2 promotor are also associated with increased inhibitory H3K9me2 and H3K9me3 marks at this promotor in endothelial cells. There is also evidence that demethylation of the NOS2 promotor allows for enhanced iNOS expression upon LPS and IFNγ treatment [119].

Protein acetylation has also been demonstrated as an important modulator of NOS2 gene transcription. Binding of NFκB to the NOS2 promoter initiates recruitment of epigenetic modifiers such as HAT enzymes (ex. p300), which acetylate NFκB subunits p65 and p50 and promote NOS2 gene expression. Interestingly, HDAC inhibitors have been shown to suppress induced NOS2 gene transcription through multiple mechanisms including acetylation at inhibitory sites or acetylation of additional subunits (i.e., p52) promoting formation of a complex that cannot bind DNA. Additionally, under conditions of high glucose concentration, Set7 methyltransferase expression is increased resulting in accumulation of H3K4me1 at the RELA gene that encodes the NFκB subunit p65 [120]. Known connections between ATM regulation of the key transcription factor involved in regulating most of the genes that become tolerized in suppressed innate immune cells of sepsis survivors further raises the possibility that ATM is involved in this epigenetic reprogramming.

### 4.5. ATM Regulation of Hematopoietic Stem Cells (HSCs)

Intriguingly, recent studies indicate roles for ATM regulation specific to the HSC population through its functions as both a redox sensor and DDR factor. One study has demonstrated that an ATM-AKT signaling pathway is triggered by cellular stressors, which can lead to induction of mitochondria-dependent apoptosis processes within HSCs [121,122]. Moreover, increased ATM activation through oxidation may lead to increased autophagy and better sepsis outcomes. Evidence is also accumulating that redox signaling is an important regulator of HSC self-renewal and mobilization [123]. ATM is essential for self-renewal of HSCs [96] and was found to regulate self-renewal and quiescence of HSCs through its antioxidant response [123]. This role was first uncovered when mice deficient in ATM (atm^−/−^) aged 24+ weeks exhibited progressive bone marrow failure due to defective HSC function associated with elevated ROS, which was rescued by treatment with the antioxidant N-acetyl-L-cysteine (NAC). The HSC dysfunction was shown to be induced by increased ROS and subsequent activation of the p16-Rb gene product pathway [96].

In its role as an effector of the DDR, ATM has been shown to phosphorylate BID, which is important in maintaining HSC quiescence. Loss of BID phosphorylation led to elevated levels of ROS, limited self-renewal capacity, and exhaustion of the HSC pool, which were rescued by NAC treatment. It is suggested that BID may induce ROS production by mitochondria, which is normally suppressed by ATM through prevention of BID accumulation within mitochondria [123]. These studies support the hypothesis that signaling through ATM may contribute to the epigenetic changes that occur in hematopoietic stem and progenitor cells (HSPCs) during sepsis.

### 4.6. ATM as a Coordinator of Epigenetic Change during Sepsis

Further evidence of ATM involvement in epigenetic changes to HSCs in response to oxidative stress and DNA damage is that ATM is a known modulator of cellular senescence, which is known to impart a distinct pattern of epigenetic changes to the chemical markers and structure of chromatin similar to that of immunosuppressed HSCs. The suppression of HSCs observed in sepsis survivors may be a form of cellular senescence. There are various known types of senescence, which can occur in different cell types and respond differently to treatment. However, cellular senescence is generally defined as a state of irreversible growth arrest that occurs in response to stress signals including RONS, cytokines, DNA damage, and activated oncogenes that results in distinct epigenetic and phenotypic changes and occurs in various disease states as well as aging.

In senescence, upregulation of the DDR results in activation of proteasomal degradation of histone methyltransferases G9a and G9a-like protein (GLP), decreasing the level of H3K9 methylation that transcriptionally repress certain senescence-associated gene promoters [124]. In addition, upon upregulation of the DDR and triggered by remodeling of chromatin structure, HDAC1 is inhibited, leading to hyperacetylation of chromatin-associated proteins and increased expression of osteopontin [125].

ATM represses DSB-induced expression of senescence-associated genes, including the genes encoding the WRKY and NAC transcription factors, which are central components of the leaf senescence process, via modulation of histone lysine methylation [126]. Additional evidence for ATM involvement in outcome of sepsis includes two key findings showing that (i) the PARP1 inhibitor Olaparib confers beneficial effects to mice subjected to CLP sepsis [127], and (ii) anthracyclines trigger ATM activation, which is associated with 80% improvement in septic mouse survival [128]. Both PARP1 and ATM are major regulators of the DDR [129]. Additionally, the interplay between activation of the DDR and the innate immune response during infection is due to the major roles PARP1 and ATM play in innate immune cell homeostasis [41]. The dual roles of ATM in DNA repair and epigenetic regulation are summarized in Figure 3.

## 5. Involvement of Additional DDR Factors in Coordinating Epigenetic Changes

There is evidence that additional proteins involved in coordinating the DDR could also be involved in the epigenetic reprogramming of HSCs. In particular, recent evidence indicates that p53 and p21 are master regulatory proteins involved in initiating senescence of HSCs. Characteristic markers of the senescence associated secretory phenotype (SASP) can include enlargement of cellular morphology, increased lysosomal β-galactosidase activity, and increased p16 expression [130].

The two most widely accepted pathways of senescence initiation include DDR-dependent and -independent mechanisms, which both involve upstream orchestration by p53. p53 is referred to as the “guardian of the genome” since its responses mediate whether cells will enter a DDR, apoptotic, necrotic, or senescence pathway to protect genomic integrity. The two most extensively studied pathways that mediate senescence in response to DNA damage are the (i) p53/p21^Cip1/Waf1^ and (ii) p16^INK4A^/Rb pathways. Detection of DNA damage leads to (i) activation of p16^INK4A^, which inactivates CDK4/6 and leads to accumulation of phosphorylated Rb, subsequently inhibiting activity of E2F transcription factors or (ii) activation of ATM or ATR, which phosphorylate and activate Chk2 or Chk1, respectively, leading to p53 activation and upregulation of p21, both of which result in cellular senescence [131]. The mechanism through which p53 induces senescence independently of DDR activation occurs through the phosphatase and tensin homolog (PTEN)-loss-induced cellular senescence pathway [132]. Depletion of PTEN, a negative regulator of the PI3K/AKT/mTOR pathway, leads to mTOR phosphorylation of p53 at serine Ser15. mTORC1 and mTORC2 then compete with MDM2 causing p53 accumulation, which leads to accumulation of downstream transcriptional target p21. Persistently upregulated levels of the cell cycle inhibitor p21 induces the senescent state of the cell [132,133,134,135].

Cellular senescence is associated with particular epigenetic modifications and changes to chromatin structure that are thought to be mechanistically involved in inducing the senescent phenotype [136]. One distinct epigenetic change is the formation of facultative heterochromatin domains called senescence-associated heterochromatic foci (SAHF) [137] that are enriched in hypoacetylation of histones, histone H3 lysine methylation (H3K9me3 and H3K27me3), heterochromatin protein 1 (HP1) proteins, and macroH2A. These markers repress transcription of certain genes and can recruit chromatin remodeling factors such as ATRX [138]. Nuclear pericentric satellite DNA has also been shown to undergo dynamic decondensation in a phenotype termed senescence-associated distention of satellites (SADS) [139]. Coordination of the epigenetic changes that occur in senescent cells is initiated by factors such as high mobility group (HMG) protein family members [140,141], bromodomain and extra-terminal domain (BET) proteins (i.e., BRD4) [142], HDACs, and is negatively regulated by SIRT1 [143].

Lastly, it has been shown that suppression of murine BM HSCs through chemotherapy and/or IR irradiation is regulated by p53 activation, subsequent p21 upregulation, and downstream activation of p16 and p19 (p14^Arf^ in human) [144]. In this model, myelosuppression of HSCs was characterized by sustained defects in self-replication and SA-β-gal staining. Factor p16 was shown to be particularly important for maintenance of HSC senescence and resultant HSCs in the bone marrow environment experience defects in the ability to self-renew and subsequent decrease in HSC pool reserves [145].

## 6. Conclusions

In response to elevated oxidative stress that exceeds the antioxidant capacity of the cell and subsequent DNA damage in inflammatory conditions such as sepsis, persistent epigenetic changes occur to HSCs that result in immune deficiencies and increased risk of mortality of survivors. Factors involved in the DDR, such as ATM and p53, are involved in detecting elevated oxidative stress and also responding to DNA damage that occurs from production of RONS. Evidence has accumulated that ATM modulates the activity of several transcription factors with known epigenetic regulatory functions, such as HIF1α and NFκB. In particular, NFκB is the major transcription factor activated in the acute inflammatory phase of sepsis that upregulates the pro-inflammatory cytokines and effectors (i.e., TNFα, IL-1β, IL-6, iNOS, etc.) that are suppressed in tolerized innate immune cells, and it has also been shown to reprogram the chromatin (i.e., DNA and histone) methylome by regulating expression of certain histone methylases. Genes such as iNOS are known to be epigenetically suppressed through methylation of the promoter in tolerized macrophages. Additionally, ATM is also known to be important in maintenance of HSC quiescence through its role as a redox sensor and coordinator of cellular response to RONS. Collectively, these findings indicate that ATM may be a crucial moderator of the cellular response to oxidative stress and DNA damage, coordinating a response that involves epigenetic silencing of key proinflammatory genes in HSCs and resulting in hyporesponsive immune cells in survivors of inflammatory conditions such as sepsis.

Additional proteins involved in the DDR may also be involved in orchestrating epigenetic changes to HSCs in response to oxidative stress, including p53 and p21. These factors are known to be important in initiation of cellular senescence, which is a state of suppressed self-renewal capacity observed in different cell types and that may include the suppressed state observed in HSCs of sepsis survivors. Cellular senescence is characterized by a distinct pattern of epigenetic changes, which could be induced by downstream signaling of p53. Suppression of BM HSCs was in fact shown to occur through p53 activation. Clearly, systemic inflammation occurring during severe sepsis is highly complex and inflammatory signaling pathways outside of RONS or the DDR likely also contribute to epigenetic changes occurring in sepsis.

Taken together, recent evidence suggests that components of the DDR, such as ATM and p53, may play critical roles in the cellular antioxidant response to elevated oxidative stress and inflammation through epigenetic regulation of HSCs to impart persistent changes in immune responses.

## Figures and Tables

**Figure 1 antioxidants-10-01146-f001:**
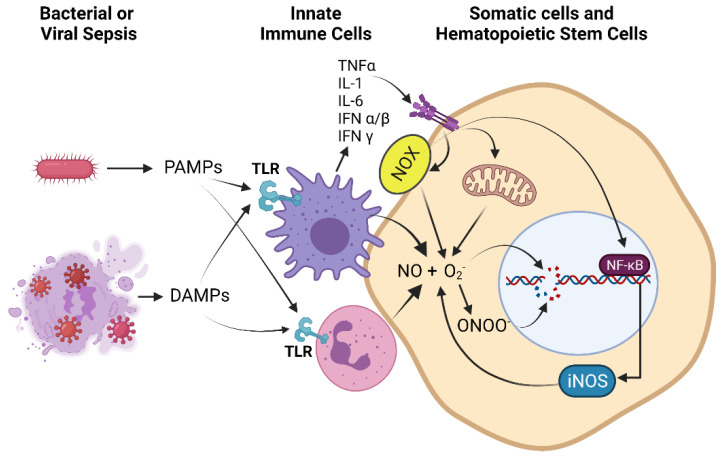
Immune cell activation during the hyper-inflammatory phase of sepsis, secretion and amplification of cytokines, and up-regulation of RONS production by immune, somatic, and hematopoietic stem cells: Pathogen-associated molecular patterns (PAMPs) and danger-associated molecular patterns (DAMPs) generated by bacterial or viral sepsis activate pattern recognition receptors (PRRs) such as toll-like receptors (TLRs) of innate immune cells such as macrophages and neutrophils (shown), leading to ROS and RNS production as part of the pathogen killing response; however, these cells also adhere to vascular and parenchymal cells causing oxidant injury [71]. In addition, there is upregulation and secretion of pro-inflammatory cytokines (cytokine storm) such as tumor necrosis factor α (TNFα), interleukin-1 (IL-1), interleukin-6 (IL-6), interferon α/β (IFNα/β), and interferon γ (IFNγ) from activated macrophages. Proinflammatory signaling leads to upregulation of NADPH oxidase (NOX), which produces reactive oxygen species (ROS) such as superoxide (O_2_^−^), and upregulation of inducible nitric oxide synthase (iNOS), which produces nitric oxide (NO), in immune, somatic, and hematopoietic stem cells. NO and O_2_^−^ can combine to produce peroxynitrite (ONOO^−^). Elevated levels of these reactive oxygen and nitrogen species (RONS) lead to DNA damage and upregulation of the DNA damage response (DDR). Additional ROS such as H_2_O_2_, hydroxyl radical and hypochlorous acid also contribute to DNA damage but are not shown for clarity of presentation.

**Figure 2 antioxidants-10-01146-f002:**
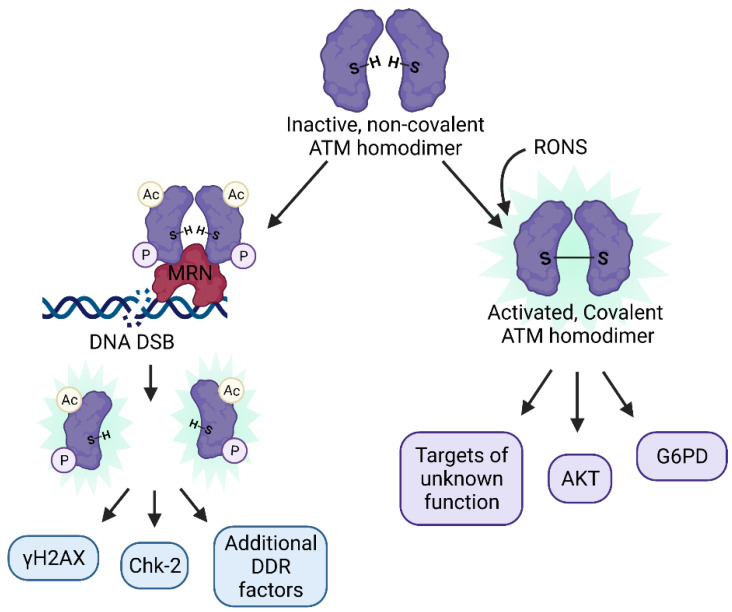
The two distinct mechanisms of ATM activation: ATM usually exists in an inactive, non-covalent homodimer form that can become activated via (left) the canonical DDR upon recruitment to the MRN complex near the site of DSBs, trans autophosphorylation and acetylation, monomerization, and targeting of downstream DDR effectors such as H2AX, Chk-2, KAP-1, p53, and others, or (right) through oxidation by RONS of cysteine residues that result in a covalently attached ATM homodimer, which targets AKT, G6PD, and hundreds of additional downstream substrates, many of which are of unknown function.

**Figure 3 antioxidants-10-01146-f003:**
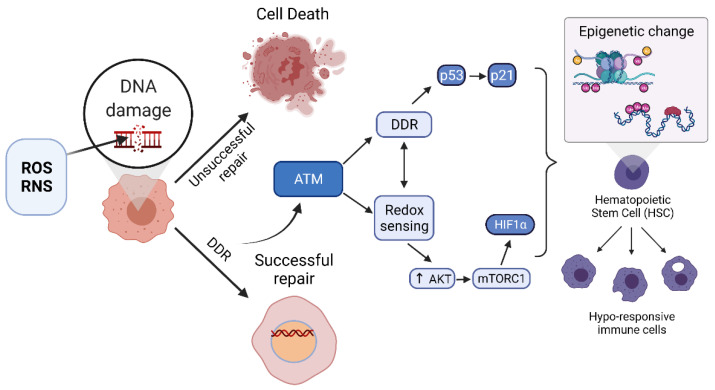
Successful upregulation of the DNA damage response (DDR) including high-fidelity repair responses activate ATM, which can activate additional DDR factors and can separately become activated by direct oxidation in a redox sensing manner. Activation of ATM leads to upregulation of transcription factors and epigenetic processes in somatic as well as hematopoietic stem cells (HSCs) that lead to suppression of pro-inflammatory programs in progeny innate immune cells.

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
