# Peer review of "Mechanisms of Ataxia Telangiectasia Mutated (ATM) Control in the DNA Damage Response to Oxidative Stress, Epigenetic Regulation, and Persistent Innate Immune Suppression Following Sepsis"

_antioxidants, 2021, doi:10.3390/antiox10071146_

Round 1
Reviewer 1 Report
This work includes a literature review of oxidative factors that play an important role in sepsis as a model of the pathophysiological state of inflammation that causes increased oxidative stress, DDR upregulation, and epigenetic reprogramming of hematopoietic stem cells (HSCs). The authors conclude that, in light of recent evidence, the components of DDR , such as ATM and p53, may play a key role in the cellular antioxidant response to elevated oxidative stress and inflammation by epigenetic regulation of HSC to induce permanent changes in immune responses. The manuscript is basically ready for publication. However, I have a suggestion for authors to consider shortening the title of the article, it is too long and may mislead the reader.
Author Response
Point 1:
This work includes a literature review of oxidative factors that play an important role in sepsis as a model of the pathophysiological state of inflammation that causes increased oxidative stress, DDR upregulation, and epigenetic reprogramming of hematopoietic stem cells (HSCs). The authors conclude that, in light of recent evidence, the components of DDR , such as ATM and p53, may play a key role in the cellular antioxidant response to elevated oxidative stress and inflammation by epigenetic regulation of HSC to induce permanent changes in immune responses. The manuscript is basically ready for publication. However, I have a suggestion for authors to consider shortening the title of the article, it is too long and may mislead the reader.
Response 1:
The reviewer indicated that the manuscript is basically ready for publication. Thank you. They did recommend revising the title. We have done so to make it shorter and more focused on the content of the review.
Reviewer 2 Report
Sepsis is a life-threatening condition that requires immediate and intensive treatment. Recovering from its acute stage, patients can develop significant and lasting immunosuppression, which resembles an immune response deficiency associated with senescence or cancer progression. In the present manuscript, the authors provide a comprehensive overview of sepsis-induced immunodeficiency and aggregate data on the potential role of ATM kinase in its development. Their review details the effects of reactive oxygen species (ROS) and resulting DNA damage. The Authors note that substantially different levels of ROS are involved in physiological signaling, stress signaling, and ROS-induced damage. Increased oxidative stress is observed in sepsis. Also, markers of DNA damage were found in these conditions. However, it remains unclear to me how profound this effect is? What part of cellular populations can undergo stress-related epigenetic reprogramming?
Provided figures are informative. However, I found Figure 1 somewhat confusing and not well coupled with the text. The source of ROS and NO is unclear. The current figure might suggest that innate immune cells secrete ROS, which other cells internalize to suffer damage. Two different ‘innate immune cells’ with apparently different functions are not labeled. On the other hand, Figure 1 introduces ‘danger-associated molecular patterns (DAMPs)’, which are not referred to in the manuscript text. Moreover, Figure one is first referred to depict ‘cytokine storm’ (see line 288), which is not directly mentioned on the figure or its legend.
Also, Graphical Abstract, which shares its part with Figure 3, is not self-explanatory and not easy to follow. In my opinion, it would benefit from simplification or, if allowed by journal rules, from the addition of a legend.
The senior authors are experts in the field. The manuscript sufficiently covers significant aspects of this still-developing area. A somewhat weak point of the manuscript is the relatively frequent referral to other reviews in place of the original findings. Yet, the text is adequate and provides a useful resource. As the review aims to provide potential links of ATM and immunosuppression, it is somewhat imbalanced, which is justified in this case. Still, the concluding part could point to other open directions.
Overall, the review can be attractive to a broad range of readers. Below are a few minor comments:
Figure 1 would benefit from a caption.
Mark G. Clemens is marked as the corresponding author, but, email addresses of Laura A. Huff and Shan Yan are provided instead (lines 10-11).
Author Response
Point 1:
Sepsis is a life-threatening condition that requires immediate and intensive treatment. Recovering from its acute stage, patients can develop significant and lasting immunosuppression, which resembles an immune response deficiency associated with senescence or cancer progression. In the present manuscript, the authors provide a comprehensive overview of sepsis-induced immunodeficiency and aggregate data on the potential role of ATM kinase in its development. Their review details the effects of reactive oxygen species (ROS) and resulting DNA damage. The Authors note that substantially different levels of ROS are involved in physiological signaling, stress signaling, and ROS-induced damage. Increased oxidative stress is observed in sepsis. Also, markers of DNA damage were found in these conditions. However, it remains unclear to me how profound this effect is? What part of cellular populations can undergo stress-related epigenetic reprogramming?
Response 1:
This reviewer indicated that it is unclear how widespread oxidative stress is in sepsis and what cellular populations are involved. We have clarified this on p. 7, lines 293 – 302 with additional information and an additional reference in the legend for figure 1.
Point 2:
Provided figures are informative. However, I found Figure 1 somewhat confusing and not well coupled with the text. The source of ROS and NO is unclear. The current figure might suggest that innate immune cells secrete ROS, which other cells internalize to suffer damage. Two different ‘innate immune cells’ with apparently different functions are not labeled. On the other hand, Figure 1 introduces ‘danger-associated molecular patterns (DAMPs)’, which are not referred to in the manuscript text. Moreover, Figure one is first referred to depict ‘cytokine storm’ (see line 288), which is not directly mentioned on the figure or its legend.
Response 2:
The reviewer also found figure 1 somewhat confusing. We have now revised the figure and its legend to clarify which inflammatory cells are involved and that they deliver ROS and NO primarily on adhesion. We also clarified that the cytokines depicted in the figure are the “cytokine storm) and address DAMPS and PAMPS in the text (lines 287-289).
Point 3:
Also, Graphical Abstract, which shares its part with Figure 3, is not self-explanatory and not easy to follow. In my opinion, it would benefit from simplification or, if allowed by journal rules, from the addition of a legend.
Response 3:
The reviewer also suggested that the graphical abstract be simplified for clarity. We have revised it accordingly. Unfortunately, a figure legend is not allowed for the Graphical Abstract.
Point 4:
The senior authors are experts in the field. The manuscript sufficiently covers significant aspects of this still-developing area. A somewhat weak point of the manuscript is the relatively frequent referral to other reviews in place of the original findings. Yet, the text is adequate and provides a useful resource. As the review aims to provide potential links of ATM and immunosuppression, it is somewhat imbalanced, which is justified in this case. Still, the concluding part could point to other open directions.
Response 4:
We agree that while the “somewhat imbalanced” focus on ATM pathways is justified, other mechanisms should be acknowledged (p. 18, lines 729 -731).
Point 5:
Overall, the review can be attractive to a broad range of readers. Below are a few minor comments:
Response 5:
Thank you for this comment.
Point 6:
Figure 1 would benefit from a caption.
Response 6:
The reviewer suggested that figure 1 could benefit from a caption. One is provided in the original submission. It is possible that the reviewer is referring to the graphical abstract. We mistakenly included that in the manuscript body as well as uploading as a separate file. We have deleted it from the body of the manuscript.
Point 7:
Mark G. Clemens is marked as the corresponding author, but, email addresses of Laura A. Huff and Shan Yan are provided instead (lines 10-11).
Response 7:
We have also corrected the email address for correspondence.